# Retinoic Acid and Its Derivatives in Skin

**DOI:** 10.3390/cells9122660

**Published:** 2020-12-11

**Authors:** Łukasz Szymański, Rafał Skopek, Małgorzata Palusińska, Tino Schenk, Sven Stengel, Sławomir Lewicki, Leszek Kraj, Paweł Kamiński, Arthur Zelent

**Affiliations:** 1Department of Molecular Biology, Institute of Genetics and Animal Biotechnology, Polish Academy of Science, Postępu 36A, 05-552 Magdalenka, Poland; l.szymanski@igbzpan.pl (Ł.S.); r.skopek@igbzpan.pl (R.S.); m.palusinska@igbzpan.pl (M.P.); 2Department of Hematology/Oncology, Clinic of Internal Medicine II, Jena University Hospital, 07747 Jena, Germany; tinoschenk@googlemail.com; 3Institute of Molecular Cell Biology, Center for Molecular Biomedicine Jena (CMB), Jena University Hospital, 07747 Jena, Germany; 4Department of Internal Medicine IV, Division of Gastroenterology, Hepatology and Infectious Disease, Jena University Hospital, Friedrich Schiller University of Jena, 07747 Jena, Germany; sven.stengel@med.uni-jena.de; 5Department of Regenerative Medicine and Cell Biology, Military Institute of Hygiene and Epidemiology, 01-163 Warsaw, Poland; 6Department of Medicine, Faculty of Medical Sciences and Health Sciences, Kazimierz Pulaski University of Technology and Humanities, 26-600 Radom, Poland; 7Department of Oncology, Medical University of Warsaw, 01-163 Warsaw, Poland; l.kraj@igbzpan.pl; 8Department of Gynecology and Oncological Gynecology, Military Institute of Medicine, 01-163 Warsaw, Poland; pkaminski@wim.mil.pl

**Keywords:** vitamin A, all-trans-retinoic acid, retinoic acid receptors, dermatology

## Abstract

The retinoids are a group of compounds including vitamin A and its active metabolite all-trans-retinoic acid (ATRA). Retinoids regulate a variety of physiological functions in multiple organ systems, are essential for normal immune competence, and are involved in the regulation of cell growth and differentiation. Vitamin A derivatives have held promise in cancer treatment and ATRA is used in differentiation therapy of acute promyelocytic leukemia (APL). ATRA and other retinoids have also been successfully applied in a variety of dermatological conditions such as skin cancer, psoriasis, acne, and ichthyosis. Moreover, modulation of retinoic acid receptors and retinoid X (or rexinoid) receptors function may affect dermal cells. The studies using complex genetic models with various combinations of retinoic acid receptors (RARs) and retinoid X (or rexinoid) receptors (RXRs) indicate that retinoic acid and its derivatives have therapeutic potential for a variety of serious dermatological disorders including some malignant conditions. Here, we provide a synopsis of the main advances in understanding the role of ATRA and its receptors in dermatology.

## 1. Introduction

Retinoids are defined as synthetic or natural derivatives of vitamin A that were first discovered in 1913. Retinol and retinyl ester are dietary forms of what is commonly known as vitamin A. These forms of vitamin A are not biologically active and require transformation, by cytosolic alcohol dehydrogenases (ADHs) and microsomal retinol dehydrogenases (MDHs), to become retinaldehyde, and subsequent oxidation by retinaldehyde dehydrogenases RALDH1, RALDH2, and RALDH3 to retinoic acid (RA) [1]. While the reverse transformation of retinaldehyde to retinol is possible through the DHRS3 enzyme activity, the retinaldehyde to RA conversion is irreversible [2]. RA exists in several isoforms of which the most common are all-trans retinoic acid and 9-*cis* retinoic acid [2]. All-trans-retinoic acid (ATRA) is also one of the main physiologically active metabolites of vitamin A. Retinoids, which are hydrophobic compounds, require retinoid-binding proteins for stabilization in aqueous media. Depending on the localization, the stabilization of retinoids is achieved by binding to different proteins such as cellular retinol-binding proteins (CRBPs) or cellular retinoic acid-binding proteins (CRABPs), the interstitial retinol-binding protein (RBP 3), and plasma-retinol binding protein (RBP 4) [3]. The half-life of RA is around 1 h because it is rapidly metabolized by the cytochrome P450 enzymes (CYP26s) [2]. CYPs are involved in ATRA hydroxylation and thus inactivation of its function. RAs are not only substrates for CYP26 enzymes but are also potent *CYP26* inducers, therefore creating a negative feedback loop [4].

It is believed that ATRA deficiency, mediated by CYPs or other mechanisms, is associated with cancer progression and various dermatological diseases [5].

Retinoids are usually classified in one of the three generations; however, some researchers consider pyranones derivatives as the fourth generation. Briefly, naturally occurring, non-aromatic retinoids are classified as the first generation, monoaromatic vitamin A derivatives are the second generation, whereas retinoids containing a cyclic polyene side-chain are the third generation [3]. More detailed information about retinoic acid and its signaling pathways may be found in a recent review by Ghyselinck and Duester (2019) [2].

## 2. Retinoic Acid Receptors and Molecular Mechanism of Their Action

Retinoic acid receptors are key developmental regulators and, as such, function as a molecular switch in many developmental processes including skin development. In the absence of a ligand, RARs repress transcription through recruiting histone deacetylases (HDAC) complexes such as HDAC-N-CoR (negative co-regulator) or HDAC-SMRT (silencing mediator for retinoid and thyroid hormone receptors) (Figure 1) [6].

In the presence of a ligand, change in the position of helix 12 occurs and co-repressors are replaced with co-activators such as DRIP/TRAP/ARC (Vitamin D3 receptor-interacting proteins/Thyroid hormone receptor-associated proteins/Activator-recruited complex) which leads to gene-targeted transcription activation through chromatin decompression. Three different RAR coding genes (-α, -β, and -γ) and retinoid X (or rexinoid) receptor (RXR) genes have been characterized. Each of these genes encodes multiple N-terminal protein isoforms (differing in their N-terminal regions) which can be produced by differential promoter usage and alternative splicing [7]. RXRs serve as the obligatory heterodimerization partners for RARs and several other nuclear receptors including those for thyroid hormones and vitamin D3, thus integrating different signaling pathways [8]. Like other *RAR* genes, RARα encodes two major isoforms: RARα1 and RARα2. These isoforms differ in their A regions that are generated by alternative splicing and differential promoter usage and are identical in their B to F region sequences, which contain a ligand (LBD) and a DNA (DBD) binding domain. The isoforms are also identical in terms of structural motifs responsible for dimerization, co-repressor interaction, and ligand-dependent *trans*-activation. The difference in the A regions contributes to different transcriptional regulation in a cell promoter-specific and ligand-independent manner [9]. Expression of the isoforms RARα2, RARβ2, and RARγ2 is induced by ATRA and is controlled by the promoters containing RAREs (retinoic acid response elements) sequences. RARs dependent transcription activation is intrinsically linked to their proteasome-mediated degradation. At the same time, ATRA upregulates the expression of RARα2, RARβ2, and RARγ2 isoforms, and induces proteasome-dependent degradation of retinoic acid receptor, which is related to this ubiquitination as described for many other proteins. Therefore, it might be possible that activation of transcription by RARs related to proteasome-mediated degradation and upregulation of RARs expression by ATRA was evolutionarily favored to sustain the expression of a given receptor and maintain gene regulation and physiological effects of ATRA over an extended time [10].

Physiologically, to interact with its nuclear receptor, ATRA is transported into the nucleus by the CRABP that binds ATRA with high affinity. CRABP are divided into two subtypes, CRABP I, and CRABP II that is more abundant in the skin [11]. CRABP II has been proposed to be a marker of RA activity in human skin. Expression of CRABP II is induced by ATRA and reduced in aging human and mouse skin. Research suggests a role of CRABP II in skin aging. Knock-out of the *CRABP II* gene in mice causes, among other things, the reduction of keratinocyte layers, degree of proliferation and differentiation, and skin thickness. Loss of the *CRABP II* gene leads to the reduction and loosening of collagen bundles, which contributes to premature and severe skin aging [12].

RARα, RARγ, and RXR are highly expressed in the fibroblasts and keratinocytes with RARα being more abundant in the fibroblasts. RARβ is either not present or expressed at a low level; however, exposure to retinoic acid rapidly induces its expression [13]. In the human epidermis, the abundance of the given type of heterodimer complex depends on the keratinocyte differentiation state, and so the RARα/RXRα complex is dominant in the basal layer whereas the RARγ/RXRα complex is most common in the suprabasal layer [14].

Upon binding of ATRA to the LBD, the RARs exhibit a conformational change allowing for heterodimerization with an RXR. However, the RAR is not an exclusive heterodimerization partner for RXR. Thyroid hormone receptor, estrogen receptor, constitutive androstane receptor, vitamin D receptor, and many other nuclear receptors can act as a heterodimeric partner for RXR. Therefore, the broad range of RA effects (through the 9-*cis* RA) is not induced solely by the RAR/RXR mechanism of action but also by the physiological effects of activation of any RXR heterodimerization partners [7]. ATRA and its isomers, 9-*cis*-RA and 13-*cis*-RA, are the naturally occurring ligands for the RARs. They can bind three RARs with different affinities. The 9-*cis*-RA is a high-affinity ligand for RXR, and a selective agonist of RARα, RARβ, and RARγ. The RARα is bound with the highest affinity by the 9-*cis*-RA followed by the 13-*cis*-RA and ATRA, whereas for RARβ and RARγ, the order of affinity is reversed [15,16]. A detailed description of the retinoic acid receptors and their functions is presented in Das et al. [17].

## 3. Non-Genomic Effects of ATRA

Like other nuclear receptor ligands, ATRA can act independently of the classical mechanism of nuclear receptor action through the non-genomic pathway that does not require gene transcription (Figure 2).

ATRA and other retinoids activate several cellular kinases in a highly cell-specific manner. For example, the p38 mitogen-activated protein kinase (p38MAPK) is noncanonically activated in various cells such as leukemia cells, mouse embryonic carcinoma cells, and fibroblasts [18]. Phosphatidylinositol-3-kinase (PI3K) and extracellular signal-regulated kinase 1/2 (ERK1/2) can also be modulated by ATRA through the non-genomic pathway [19,20]. Although the exact mechanisms underlying the non-genomic effects of ATRA are not fully understood, it is clear that these effects have important consequences and have to be taken into consideration when evaluating the effects of retinoids on various types of cells. In this respect, it is worth noting that ATRA, through the activation of multiple kinase signaling pathways, can also exert transcriptional effects independent of RARs via transcription factors that lie at the end of these signaling cascades. ATRA can also lead to the up-regulation of G-CSF and GM-CSF receptor expression, thus resulting in the amplification of a particular signaling cascade via a positive feedback loop mechanism [21,22,23]. Likewise, ATRA activity in skin cells will depend on complex interactions between genomic and non-genomic pathways that remain to be fully understood.

## 4. Retinoic Acid in the Skin

Retinoids, besides their success in APL (acute promyelocytic leukemia) therapy, are also widely used in the treatment of skin diseases such as skin cancer, psoriasis, acne, ichthyosis, and even wrinkles because of their effects on cell differentiation, proliferation, and apoptosis [24,25,26]. The usefulness of ATRA in the skin appears to be partially limited due to ATRA-mediated resistance caused by a variety of multifactorial mechanisms. The reason that ATRA has limited efficacy in the clinics may be linked to CYPs activity. Since RAs act towards normalization of the skin by regulating the keratin’s expression, an unregulated degradation of ATRA by CYPs can cause a RA deficiency state related to the progression of hyperkeratinization, desquamation in the context of acne, psoriasis, and ichthyosis [14,27,28]. To overcome these limitations, novel strategies associated with exogenous ATRA therapy have been proposed. These strategies are based on modulation and increasing levels of endogenous ATRA by inhibiting ATRA-4-hydroxylase enzymes, which are responsible for ATRA metabolism. These inhibitors are also referred to as retinoic acid metabolism-blocking agents (RAMBAs) and they are considered crucial in ATRA-mediated therapy [29]. The main mechanism of RAMBAs action is associated with inhibition or blocking of 4-hydroxylation of all-trans-retinoic acids, which depend on cytochrome P450-dependent enzymes (CYPs). This in turn results in an increase of the intracellular all-trans-retinoic acids [30]. Nowadays, there are a large number of RAMBAs which may modulate the activity of ATRA. This blocking aging may be divided into several groups of compounds which are structurally or chemically similar, i.e., to liarozole (LiazalTM) and related compounds—R115866 and R116010; azolyl retinoids and related compounds, benzene acetic acid derivatives, 2,6-disubstituted naphthalenes, or miscellaneous structures [28]. Regulation of endogenous ATRA concentration and its natural stereoisomers with the use of new RAMBAs may bring out additional cancer therapy strategies and treatments of dermatological diseases [28]. Ketoconazole (from azole) was the first RAMAB compound described by Van Wauwe et al. in 1988 [31]. In high doses (400 mg three times a day), ketoconazole affected the androgenic hormone, which was used in the treatment of prostatic cancer. In dermatology, the drug is used mainly as an antifungal agent used for the treatment of superficial and systemic fungal infections. The most studied RAMBA compound is liarozole. It has been proven that liarozole exhibits anti-tumor properties against prostate and breast cancers [32]. In dermatology, liarozole is used in treatment of psoriasis and ichthyosis [33]. In RAMBAs’ history, there were also examples of very promising but not widely used drugs. One of them—talarazol—was carefully investigated in the past decade for the treatment of acne, psoriasis, and other keratinization disorders; however, despite its properties, it was withdrawn from the market after the results of clinical trials [30].

## 5. Skin Differentiation

Retinoic acid strongly influences epithelial differentiation and proliferation. In vitro cultured keratinocytes in media lacking retinoic acid have reduced motility, do not form characteristic patterns, and have increased adhesiveness. What is more, the production of keratins is regulated by the concentration of retinoids. In cells cultured without retinoic acid, the synthesis of keratin 1 (67 kDa), which is characteristic for terminally differentiating keratinocytes, is increased, while the synthesis of keratin 8/18 (52 kDa) and keratin 19 (40 kDa) is decreased [34]. On the other hand, supplementation of retinoids in cultured human epidermal keratinocytes leads to increased expression of keratin 7, keratin 13, keratin 15, and keratin 19 and reduced expression of keratin 1, keratin 5, keratin 6, keratin 10, and keratin 14 [14]. On the molecular level, ATRA regulates more than 3000 genes in keratinocytes and induces changes as soon as 1 h after addition to cells in culture. Most of the affected genes are involved in the regulation of DNA synthesis and repair, cell cycle, translation, adhesion, transcription factors, RNA metabolism, apoptosis, receptor expression, protein kinase, and membrane proteins. Interestingly, retinoic acid controls its own bioavailability by regulating ATRA synthesis and metabolism. It was shown that retinoic acid downregulates the expression of RARγ and upregulates the expression of RARβ (48 h and 72 h after ATRA exposure, respectively), while RARα is not regulated by ATRA in vitro [25]. Saitou et al., 1995 [35], showed, using a mouse model, that the inhibition of RAR-dependent signaling in the basal cell layer leads to immature epidermis formation by downregulation of keratinocyte differentiation. Retinoic acid also proved to be essential in preadipocytes differentiation in adipose tissues [36,37].

## 6. Epidermal Barrier

Out of the layers of the epidermis, the stratum corneum plays a major role in the formation of the epidermal barrier. The stratum corneum consists of 15–20 layers of corneocytes, which are keratinocytes with cornified envelopes but lacking cytoplasm and nucleus, intracellular lipid lamellae, and corneodesmosomes. Due to its structure, the stratum corneum is a primary shield that protects the body from the environment; however, the internal and external factors involved in cell adhesion, differentiation, and proliferation can impair epidermal barrier integrity [38].

To maintain its protective function, damaged skin must be repaired through the complex process called wound healing. However, if the process is disrupted by age or disease, chronic wounds may develop. Lee et al., 2020 [39], showed in Human Skin Equivalents that Seletinod G, a synthetic retinoid, facilitates the wound-healing process by promoting the migration of keratinocytes in the epidermis, and thus improves epidermal barrier function.

Impairment of epidermal barrier function is commonly caused by age-related flattening of the dermal-epidermal junctions and reduced proliferation of keratinocytes. Both of these processes also can be inhibited with the use of RA or retinol since both increase the mitotic activity of keratinocytes and therefore enhance skin cell proliferation as observed by epidermal thickening. What is more, Kong et al., 2016 [40], observed in histological samples that retinol and RA treatment reversed the flattening of dermal-epidermal junctions and induced the development of rete ridges.

On the other hand, RA affects the differentiation process in keratinocytes, which leads to the suppression of genes involved in the biosynthesis of epidermal lipids. Lipids synthesized by the keratinocytes are necessary for the proper cohesion of the stratum corneum and therefore for the maintenance of epidermal barrier function [25,41]. Finally, Li et al., 2019 [42], showed in a mouse model and in the HaCaT cell line that ATRA influences the expression of genes associated with barrier dysfunction, proteases, cornfield envelope, and tight junctions. The authors observed that ATRA altered the localization and downregulated the expression of Claudin-1, a key protein involved in tight junctions, and thus in epidermal barrier function.

## 7. New Strategies in ATRA and Other Retinoids-Mediated Therapy of Skin Conditions

Many therapies have been proven to be effective in the treatment of skin conditions derived from the depletion of endogenous retinoid concentration.

In the research of Cohen et al., 2001 [43], ATRA was successfully administered to patients with multiple miliary osteoma cutis. Local administration of ATRA caused a decrease in the number of papules on the face and even their elimination in patients. The authors suggest that this effect was achieved by encouraging the normal differentiation of fibroblasts rather than ATRA-dependent transepidermal elimination. What is more, Kong et al., 2016 [40], showed that ATRA and retinol treatment lead to upregulated expression of collagen type 1 (*COL1A1*) and collagen type 3 (*COL3A1*) genes and thus to increased procollagen I and procollagen III protein levels. Both compounds increased epidermal thickness and affected reduction in facial wrinkles after 12 weeks of therapy, contributing to the anti-aging effect.

The use of retinoids in anogenital HPV therapy was also evaluated. Daily application of 0.05% ATRA cream (tretinoin) completely removed HPV warts in 85% of 25 children compared to 32% in 25 children in the control group [44]. Oral administration of ATRA at 1 mg/kg dosage for 3 months led to complete remission in 16 subjects [45]. Retinoids stimulated desquamation and curbed proliferation of HPV-infected keratinocytes, but were ineffective in the treatment of condylomata acuminata [46,47]. It is well documented that ATRA is able to protect fibroblasts and other skin cells against UV radiation-induced oxidative damage. A study conducted by Cheng et al., 2019 [48], concluded that ATRA significantly induces Nrf2 (nuclear factor erythroid 2–related factor 2). E3 ligase Hrd1 induces Nrf2 ubiquitylation and degradation in embryonic fibroblast cells in response to oxidative stress, while Nrf2 acts as an ATRA-activated photoprotective agent in the skin regeneration. In the mentioned research, ATRA reversed upregulation of E3 ligase Hrd1 expression and upregulation of Nrf2 expression in UV-exposed cells both in vivo and in vitro [49].

Li et al., 2017 [50], also evaluated the RAR- and RXR-dependent mechanism of the therapeutic effects of ATRA on photoaged skin in mice models. The study showed that ATRA and RAR agonists limited the UV-induced damage to collagen fibers, and increased collagen content in photoaged skin. ATRA and RAR agonists stimulated type I procollagen protein expression and inhibited matrix metalloproteinases (MMPs) function. Inhibition occurred by downregulation of c-Jun protein and led to decreased collagen degradation in photoaged skin. ATRA also antagonized UV-mediated activation of the AP-1 transcription factor responsible for decreased procollagen synthesis. The authors concluded that ATRA and RAR agonists are able to ameliorate UV-induced damage to the skin and increase collagen content in photoaged skin through RAR. RXR was proved to have no significant effect on increased collagen content and collagen fibers [42]. Although ATRA is successfully used in the treatment of various skin conditions, its application is still given much concern due to many mild side effects of the treatment. As mentioned before, ATRA induces differentiation of skin cells but simultaneously upregulates the exfoliation of altered cells of the epidermis. Most ATRA-based treatment methods rely on the application of ATRA on the skin layer in the form of cream. Sumita et al., 2018 [51], compared the efficacy of 0.05% and 5% tretinoin cream on photoaged skin and multiple actinic keratosis. The treatment resulted in skin photoaging improvement. The thickness of the epidermis and solar elastosis were not affected by the treatments. Alterations in collagen I were also not observed. In 5% tretinoin cream treatment, there is a stronger stabilization effect on field cancerization than with 0.05% cream. Researchers suggest that dermal repair and reversal of solar elastosis may need tretinoin. A study showed that different patterns of action of tretinoin occur in photoaging skin. In patients, 0.05% cream restored the dermis, while 5% cream peeled proliferative keratinocytes. Campione et al., 2020 [52], studied a combination of 0.02% retinoic acid and 4% glycolic acid in form of a gel. Reduction in dark spots (40%) and wrinkles (12%) was observed. Overexpression of c-Jun transcription factor and TGF-β/CTGF pathway are induced by topical retinol in aged human skin in vivo [53]. Bielli et al., 2019 [12], observed that the knock-out of CRABP-II expression promoted skin aging through reduction of TGF-β signal-related genes, *Col1A1* and *Col1A2*, and an increase of *MMP2* transcripts.

Research conducted by Li et al., 2019 [42], described the effect of 0.1% ATRA cream on skin alterations in the murine model. The effect of ATRA was associated with increased exfoliation of the skin layer and, thus, its ability to remove the hyperkeratinized cells of the epidermis. The authors proved that ATRA alters the morphology and ultrastructure of the murine epidermis and plays a key role in maintaining the functional epidermal barrier. ATRA-treated skin demonstrated crust, focal parakeratosis, hyperproliferation, and intercellular edema of the stratum spinosum. What is more, infiltration of inflammatory cells was increased both with capillary dilation and increased thickness of the epidermis (2.5 higher compared to the control group). An increase in parakeratosis, spongiosis, and telangiectasia was also observed in the dermal papilla. The cytoskeletal network disappeared in the local upper stratum corneum and the keratinocyte cytoskeleton also suffered from ATRA-damaging activity. Moreover, kallikrein 6 and 10 genes were upregulated in the skin cells of mice following ATRA treatment, which may correlate with visible scales on the mouse skin. Alterations in the proteases and protease inhibitors ratio were also observed in the skin, which led to increased inflammatory response causing itching, scaling, redness, and other clinical symptoms [42].

Retinoids have been widely used in the treatment of psoriasis. This allows for a deeper understanding of the role of RAs in skin physiology and pathology. Psoriasis is a chronic, multifactorial disease during which differentiation and proliferation of keratinocytes are disturbed. The life cycle of a keratinocyte in normal skin is 26 days, while in psoriasis, it is only 4 days [54]. Hyperproliferation and aberrant differentiation of keratinocytes lead to the thickening of the epidermis, in which corneocytes still retain their nuclei. What is more, the expression of keratins is also affected and resembles these seen during wound healing. In psoriatic lesional skin, RARα and RXRα mRNA expression is significantly reduced as compared to normal skin. What is more, RARα and RARγ protein expression is increased in the lower layers of the epidermis and reduced in the upper levels when compared to normal skin, suggesting Ra’s crucial role in regulating proliferation and differentiation. Wang et al., 2020 [55], have shown in psoriasis mouse models that during the disease, the demand for vitamin A is increased and that the transformation from retinol to RA is also accelerated. They proposed that there is a positive feedback loop, in which increased demand for vitamin A caused upregulation of RBP4 and STRA6 expression and enhanced transformation of retinol to RA. In turn, RA, through binding with its receptors, stimulates the synthesis of STRA6. Since the decreased expression of retinoic acid receptors may stimulate the synthesis of STRA6, these results are in line with data presented by Laursen et al., 2015 [56]. Retinoid therapy is clinically successful in psoriasis treatment, but the underlying mechanism of action is not yet fully understood. Of more than 100 genes affected in psoriasis, around 75% are regulated by RA. As shown by Lee et al., 2009 [25], some of these genes are regulated in different directions, suggesting that RA has a “normalizing” effect on the expression of those genes. Interestingly, the authors did not observe any genes functionally related to apoptosis that are regulated by RA in psoriasis lesions even though the authors showed that RA promotes apoptosis in skin keratinocytes and that, in psoriatic skin, apoptosis is decreased [57]. This result suggests that there might exist different mechanisms of RA action not directly related to gene expression. Interestingly, RA also downregulates the vascular endothelial growth factor (*VEGF*) expression in keratinocytes, which may explain the positive effects of retinoid therapy in diseases associated with angioproliferation such as psoriasis [25]. The fact that RA does not regulate RARα in normal skin but does in psoriasis further suggests that RA has a “normalizing” effect on the gene expression.

ATRA was also proven to be effective as a fungistatic agent, especially against *Candida* and *Aspergillus fumigatus*. Therefore, it was used in the treatment of psoriasis in patients with a predisposition to fungal infections [52]. Experimental data suggested that ATRA caused a decline in the Th17 population, increased CD4þ Treg population, and promoted suppressive B cells [58].

Treatment with retinoic acid derivatives has been proven effective in basal cell carcinomas (BBCs) [59,60]. A clinical study with retinoic acid derivative—Tazarotene—was proven effective in treating BBCs. Regression of basaliomatous cells associated with reduced proliferation and increased apoptosis was observed. Unresponsive tumors showed a keratotic differentiation. Keratotic BBCs exhibited overexpression of retinol-binding protein-1 (RBP-1) and p53 compared to undifferentiated tumors. Overexpression of RBP-1 caused an increase in extracellular retinol concentration. CRBPs are essential for intracellular retinol uptake and its bioavailability. Those proteins are regulated by retinol concentration [61]. Downregulation or loss of CRBP-1 expression both with CpG island hypermethylation of CRBP-1 are connected with the progression of many tumors. Inhibition or loss of CRBP-1 may alter retinoic acid metabolism by reducing its transport, blocking RARs activity and retinyl esters formation, leading to cell differentiation inhibition and tumor progression [59], but on the other hand, it has been documented that CRBP-1 upregulation exerts a direct anti-transcriptional effect [60]. This effect is achieved by binding of the RARα to the RARE fragment of the CRBP-1 promoter [62]. RARγ and RARα upregulation is characteristic for metastatic melanoma and lung cancer [63]. Moreover, RARγ upregulation is associated with mammary tumor progression [64].

Another retinoid, the Fenretinide (4-HPR), is believed to be one of the most promising. The compound is cytotoxic to tumor cells in skin, and other cancers cells. 4-HPR affects tumors by ROS generation, increasing dihydroceramide production, promoting angiogenesis inhibition, and stimulating NK cells activity [61].

Researchers suggest that an increase in intracellular ROS levels may act as a defensive mechanism of melanoma cells against retinoic acid derivatives [65]. Orlandi et al., 2004 [60], showed that tezarotene induced concentration-dependent overexpression of RARβ associated with increased apoptosis and growth inhibition, but did not affect the expression of RARα and RARγ in basaloid tumor in vitro. Epigenetic loss of RARβ was observed in epithelial tumors and associated with tumor resistance to retinoic acid [65]. Additionally, RARα upregulation and RARγ downregulation occur in melanoma [16]. Use of 13-*cis*-retinoic acid (CRA) and HDA (histone deacetylase) inhibitors in combination reversed the epigenetic silencing of RARβ, and enhanced RA activity in RA-resistant melanoma cell lines [65]. Study conducted by Zhao et al., 2009 [66], showed that melanoma invasiveness may be suppressed by RARγ-dependent induction of carbohydrate sulfotransferase 10 (CHST10). Zhang et al., 2003 [67], examined the influence of ATRA in low (0.0001 µM) and high (10 µM) concentrations on mitochondria in melanoma. Low ATRA concentration caused a depletion in the viability of melanoma cells and reduced mitochondrial enzyme activity. At high ATRA concentrations, greater apoptosis was observed in primary and metastatic melanoma cells. Authors suggest that either the death receptor pathway or mitochondria pathway might be involved in ATRA-dependent drug-induced apoptosis mechanism [67].

Since treatment methods that rely on the application of ATRA on the skin have been proven effective, different new methods focused on increased penetration of ATRA through the skin are being investigated. Yamaguchi et al., 2005 [68], described a new delivery system for ATRA treatment in photo-damaged skin. The use of ATRA in inorganic nanoparticle coating improved permeability to the stratum corneum, reached deeper epidermis than classical ATRA treatment, and was associated with overexpression of mRNA for a heparin-binding epidermal growth factor (HB-EGF). Boost of hyaluronan among intercellular spaces of basal and spinous cell layers in the epidermis was also observed. Thus, nano-ATRA therapy efficiently influenced keratinocyte proliferation and differentiation and therefore accelerated injured skin healing. Summarized effects of ATRA on skin function are presented in Figure 3.

New drug delivery systems involving nanoparticles and nanofibers for transdermal applications were developed to deliver drugs capable of controlled release for a prolonged time [26,69,70,71,72,73,74].

One of the ATRA-containing medicines for topical administration is a hydrophilic polymer polyvinyl alcohol (PVA) covalently linked by hydrolytically degradable linkage to ATRA (PATRA). PVA as an amphiphilic nanomaterial is water soluble, and when hydrated, PATRA forms nano-fibers, which agglomerate into larger submicron-scale nanoparticles and accumulate in the skin. PATRA is characterized by increased uptake and retention in explanted pig skin compared to free ATRA. After the use of PATRA, significant reduction in gross inflammation was observed. What is more, active ATRA was released from the conjugate for 10 consecutive days in vitro [26].

A similar effect was observed in a study focused on the topical treatment of acne vulgaris using porous microparticles. The microparticles were designed to ensure a desired, sustained, and controlled release of ATRA. Microparticles with ATRA caused a significant and dose-dependent reduction in the size and thickness of inflammation in rabbit’s ears. No side effects in the form of an erythema, edema, or necrosis were observed. Microparticles in the form of a gel were also administered to the eyes of rabbits and no signs of irritation were observed, but 1/3 of studied rabbits had hyperemic blood vessels in their right eye conjunctiva. Hyperemia disappeared after 24 h, though. No symptoms at the iris or cornea were observed [73,75]. Charoenputtakhun et al., 2014 [76], described ATRA-loaded nanostructured lipid carriers (NLCs) as a tool for transdermal drug delivery. The authors claimed that NLCs are another promising dermal drug delivery system for ATRA.

Given the fact that ATRA alters skin functions and often causes side effects, it is rarely used clinically in combination with other medicines. Benzoyl peroxide is one of the compounds which exhibit a synergistic effect in combination with ATRA in acne vulgaris treatment. Although clinicians have not been eager to prescribe retinoids and benzoyl peroxide (BPO) in acne vulgaris topical treatment due to a belief that BPO may result in oxidation of ATRA and thus deplete its efficiency, it was proven that both BPO and ATRA encapsulated in microspheres exhibit a synergistic effect [69,77]. When administered topically, microspheres caused less interaction or no short-term interaction with benzoyl peroxide. Furthermore, administered synergistically, ATRA and BPO showed superior effects compared to non-encapsulated ATRA alone [70,75]. Moreover, a study conducted by Jing et al., 2017 [78] claimed that ATRA treatment resulted in a decrease in the contents of wax esters, fatty acids, and free fatty acids production. It indicated that ATRA applied topically in acne-suffering patients causes inhibition of excretion of sebum and contributed to treatment effect.

Dogra et al., 2020 [79], combined ATRA (tretinoin) with clindamycin in acne vulgaris treatment of a group of 750 patients. Combination treatment showed a 77% reduction in inflammatory, 71% non-inflammatory, and 73% total lesions and was superior to monotherapy.

Combinatory treatment with ATRA and other drugs is more effective than one drug administration, yet despite the lesser toxicity of nano-ATRA, combinational therapy should be customized to a patient’s needs.

## 8. Limitations of Use

Even though retinoids have pleiotropic and complex effects on skin that in some cases can be used to our advantage, the clinical application is limited due to the side effects. Retinoids usage leads to the build-up of tolerance and to the development of adverse effects such as erythema, rash, dryness, scaling, and desquamation, which in turn may lead to epidermal barrier dysfunction. The molecular mechanisms of the side effects of retinoids in the skin are associated with the release of proinflammatory cytokines (MCP-1, TNF-α, IL-1, IL-6, and IL-8) but is not yet fully understood, and therefore, it presents a challenge in designing new applications for RA and its derivatives [80]. What is more, concerns have been raised regarding topical retinoid administration during the first trimester of pregnancy, which may be correlated with retinoid embryopathy. Retinoid embryopathy was originally described after oral use [81]. However, there is still insufficient knowledge of retinoid embryopathy of topical treatment. A multicenter, prospective study conducted by Panchaud et al., 2012 [82], evaluated the effect of topical treatment with retinoids on pregnant women whose children were born with defects. A figure of 235 women exposed to adapalene, tretinoin, isotretinoin, motretinide, retinol, or Tazarotene constituted the study group, whereas 444 women not exposed to retinoids during pregnancy constituted the control group. The results did not suggest an increased risk of retinoid embryopathy. However, based on the current state of knowledge, topical retinoid therapy cannot be advised during pregnancy due to the possible negative effects on fetus development. Taken together, it shows that there are still a lot of gaps in the knowledge in understanding the role of retinoids in skin diseases. Future studies are necessary to complete this knowledge.

## Figures and Tables

**Figure 1 cells-09-02660-f001:**
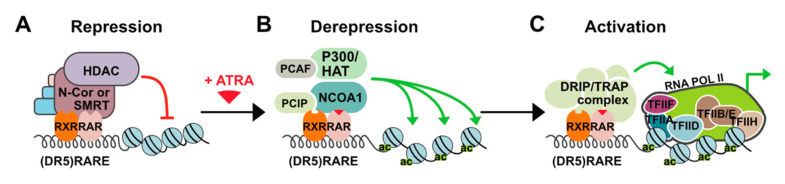
Mechanism of all-trans-retinoic acid (ATRA) action. (**A**) In the absence of a ligand, retinoid X (or rexinoid) receptors and retinoic acid receptors (RXR-RAR) heterodimers bind to retinoic acid response elements (RAREs) in the regulatory regions of target genes and repress transcription through recruiting histone deacetylases (HDAC)/co-repressor (negative co-regulator (N-CoR) or silencing mediator for retinoid and thyroid hormone receptors (SMRT)) complexes. HDACs remove acetyl groups from nucleosomal histones causing chromatin condensation that precludes binding of other factors and ultimately results in the silencing of gene expression. (**B**) Upon ligand binding, RAR undergoes structural change leading to dissociation of the co-repressor complex and association of co-activators with histone acetyltransferase (HAT) activities (e.g., P300 and NCOA1) that cause decondensation of chromatin by adding acetyl groups to nucleosomal histones. (**C**) Additional multiprotein complexes called vitamin D3 receptor-interacting proteins or thyroid hormone receptor-associated proteins (DRIP or TRAP) may also be recruited for activation of transcription through interaction with basal transcriptional.

**Figure 2 cells-09-02660-f002:**
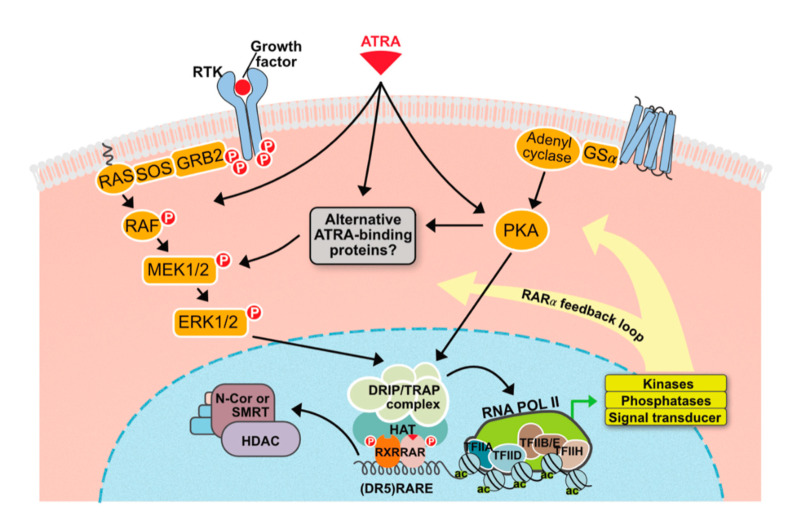
Non-genomic effects of ATRA. ATRA can cause rapid activation of various cellular kinases: shown above is the convergence of genomic and nongenomic signaling. Additionally, by directly up-regulating the expression of components of various signaling cascades, ATRA provides a positive feedback loop mechanism. Possibilities also exist for antagonistic interactions, which are not highlighted in the figure.

**Figure 3 cells-09-02660-f003:**
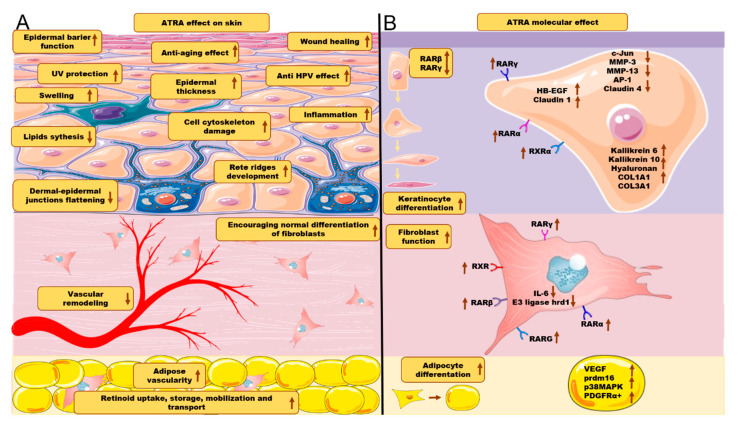
(**A**) ATRA effect on skin layers and skin cells. (**B**) ATRA molecular effects. Arrows directed upwards represent either increase or upregulation. Arrows directed downwards represent either decrease or downregulation. The figure was created using SMART (Servier Medical ART) modified graphics, licensed under a Creative Commons Attribution 3.0. Generic License.

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
