# Peer review of "Retinoic Acid and Its Derivatives in Skin"

_cells, 2020, doi:10.3390/cells9122660_

Round 1

Reviewer 1 Report

Overall this is a valuable review presenting a lot of information on recent advances in retinoic acid in dermatology.  There are some points that require improvement however.

Overall, the description of RAR function is weak.  If considered to be needed in this review then further explanation is needed for many parts (outlined below).  Or the authors could just refer to the many excellent reviews on this.  Perhaps more relevant is the description of ATRA metabolism and its regulation by ATRA as this is an important part of the discussion of ATRA action on the skin. Would also like to know more about how an “ATRA-induced CYPs which can cause an ATRA deficiency state related to the progression of hyperkeratinization, desquamation in the context of acne, psoriasis, and ichthyosis”.  How are the feedback pathways in metabolism acting in these circumstances? This would be useful to explain in the review.  More could be discussed about ATRA tolerance.

To provide comments on specific lines:

Line 49 Histone deacetylases (HDACs) are not co-repressors but an activity that some co-repressors possess, or some co-repressors may recruit HDACs

Line 53 Mediator complex is not a typical example of a co-activator for nuclear receptors as needed for very many transcription factors.  Usually discussed are earlier steps such as recruitment SRC, as shown in figure 1 (as with the co-repressors discussed earlier such as N-Cor).  Also, there is the same point as above with co-activators - histone acetylases are not co-activators but an activity that some co-activators possess or some co-activators may recruit HATs.  Abbreviations are spelled out for many proteins so should also have this for DRIP/TRAP/ARC.  This seems to be better explained in the legend for figure 1.

Line 56/57 When state “encodes multiple N-terminal protein isoforms” should explain what this means.

Line 65/66 When state “Expression of the isoforms RARa2, RARb2, and RARg2 is under the control of ATRA inducible promoters.” This means they have RAREs?  Should be more specific.

Line 66/67 The line “RARs dependent transcription activation is intrinsically linked to their proteasome-mediated degradation however, at the same time, ATRA upregulates the expression of RARa2, RARb2, and RARg2 isoforms.” Is very disjointed.  What does their  proteasome-mediated degradation have to do with ATRA regulation of receptors?  In what way is proteasome-mediated degradation important?  How does ATRA regulate the receptors?  Greatly lacking in explanation.  A few lines down please explain how this “maintains gene regulation and physiological effects of ATRA over an extended time”

Line 82/83  “ATRA must be transported by the  cellular retinoic acid-binding protein (CRABP).”  Statement “must” Is too strong.  Mouse knockouts of CRABPI and II do not have strong effects on physiology.  Further discussion of CRABPs and which are expressed in skin is relevant and should be expanded.  However, a few sentences down this section jumps back to RAR function and function of helix 12 which has already been discussed.  This is quite disorganized.   An in depth discussion of RAR function is not needed and there are many good reviews on this already. 

Line 94/95 “the broad range of ATRA effects is not induced solely by the RAR/RXR mechanism of action but also by the physiological effects of activation of any RXR heterodimerization partners [2].” Is incorrect.  Perhaps what is meant is that 9-cis RA activation of RXR may act on other partners of RXR?  This quite unclear as 9-cis RA is not mentioned.

Line 230/1 “significant interaction between ATRA and E3 Nrf2 (nuclear factor erythroid 2–related 230 factor 2).” What is meant by interaction?  They interact with each other?  Or does ATRA induce E3 Nrf2?  When Nrf2 is mentioned later is this the same as E3 Nrf2 or different?

Line 238/9 Can there be more explanation of the statement “RAR agonists stimulated type I procollagen protein expression and inhibited RAR-mediated MMP-3 and MMP-13 protein expression” It seems very odd that a RAR agonist would inhibit RAR-mediated protein expression.  Further, what actually is RAR-mediated protein expression?

Line 250 The paragraph beginning “RAR-mediated MMP-3 and MMP-13 protein expression…” – is this describing that ATRA has a positive or negative action on skin.  What is the bottom line of this paragraph?

For figure 3 why have green arrow up and red down?  The direction of the arrow is enough.  Much more useful would be to e.g. use colour to describe whether this is a positive or negative effect on the skin in disease.  In addition, the actions of ATRA are complicated – does it always induce inflammation, as shown in the figure?  Or always decrease vascular remodelling?

Grammar needs significant improvement and I would suggest that this is given a thorough check for mistakes or lack of clarity.  To pick up some of these:

Line 30/31       “Receptors” should not be plural

Line 31-33       “Complex studies showed that temporary inactivation of those receptors in skin cells indicate greater applications of retinoic acid and its derivatives in a variety of serious dermatological disorders including some malignant conditions.” I don’t understand what is meant by this sentence.  What is meant by “greater applications”. Greater than what?  Also what is meant by complex studies?

Line 47            key developmentAL regulators

Line 53            mediator complexes should be mediator complex

Line 131          These strategies ARE BASED on modulation and increasing

Line 139          do not form characteristic patternS

Line 141          retinoiC acid

Line 152/3       “(48 hours and 72 hours after exposition respectively)” what is meant by exposition?  Treatment with retinoic acid?

Line 157/8       “used in the treatment of psoriasis which allows for a better understanding of ATRAs role in the skin.” A better understanding than what method?

Line 168          “the onset demand for vitamin A is increased” what is meant by onset?

Line 183/4       “The fact that RA does not regulate RARα in normal skin but 183 does in psoriasis further indicates that RA has a “normalizing” effect on the gene expression.” Indicates is too strong and this is not sufficient to demonstrate this.  Perhaps suggests?

Line 281          “ATRA-containing medicines dedicated to topical administration” what is meant by dedicated?  Simply that it is used topically?

Line 307/8       “ATRA treatment resulted IN A decrease in the contents of wax esters, fatty acids, free fatty acids production. It indicated that ATRA APPLIED topically”

Line 313          “ Combinatory treatment with ATRA and other drugs brought out to be more effective than one drug application ordinary methods” I dont understand what is meant by ordinary methods.  A single drug?

Line 330-33     The statement “The results did not suggest an increased risk of retinoid embryopathy. Moreover, topical retinoid therapy cannot be advised during pregnancy due to the possible negative effects on fetus development.” Is confusing as the first part contradicts the second – first saying studies show no teratogenicity and then immediately after saying it possibly is a teratogen.

Author Response

Response to the Reviews

We want to thank the reviewers for all their valuable comments that allowed us to improve our manuscript. All changes of earlier text are marked in red and are included the current manuscript text according to all reviewer suggestions.  Below, we include also all reviewers’ comments with our responses.

First round  

Review 1:

Overall, the description of RAR function is weak.  If considered to be needed in this review then further explanation is needed for many parts (outlined below).  Or the authors could just refer to the many excellent reviews on this.  Perhaps more relevant is the description of ATRA metabolism and its regulation by ATRA as this is an important part of the discussion of ATRA action on the skin.

 Would also like to know more about how an “ATRA-induced CYPs which can cause an ATRA deficiency state related to the progression of hyperkeratinization, desquamation in the context of acne, psoriasis, and ichthyosis”. 

How are the feedback pathways in metabolism acting in these circumstances? This would be useful to explain in the review.  More could be discussed about ATRA tolerance.

Response: We introduced appropriate changes in the manuscript.

We improved description of RARs, ATRA, ATRA-induced CYPs, feedback pathways and ATRA tolerance. Also other sections of the paper are improved to accordance with reviewer suggestions.

Line 49 Histone deacetylases (HDACs) are not co-repressors but an activity that some co-repressors possess, or some co-repressors may recruit HDACs

Response: We changed the description: . “In the absence of ligand, RARs repress transcription through recruiting histone deacetylases (HDAC) complexes such as HDAC-N-CoR (negative co-regulator) or HDAC-SMRT (silencing mediator for retinoid and thyroid hormone receptors) (fig. 1) [6].”

Line 53 Mediator complex is not a typical example of a co-activator for nuclear receptors as needed for very many transcription factors.  Usually discussed are earlier steps such as recruitment SRC, as shown in figure 1 (as with the co-repressors discussed earlier such as N-Cor).  Also, there is the same point as above with co-activators - histone acetylases are not co-activators but an activity that some co-activators possess or some co-activators may recruit HATs.  Abbreviations are spelled out for many proteins so should also have this for DRIP/TRAP/ARC.  This seems to be better explained in the legend for figure 1.

Response: We changed the description: In the presence of ligand, change in the position of helix 12 occurs and co-repressors are replaced with co-activators such as DRIP/TRAP/ARC (Vitamin D3 receptor-interacting proteins/ Thyroid hormone receptor-associated proteins/ Activator-recruited complex) which leads to gene-targeted transcription activation through chromatin decompression.

Line 56/57 When state “encodes multiple N-terminal protein isoforms” should explain what this means.

Response: We added the explanation: Each of these genes encodes multiple N-terminal protein isoforms (differing in their N-terminal regions) which can be produced by differential promoter usage and alternative splicing [7].

Line 65/66 When state “Expression of the isoforms RARa2, RARb2, and RARg2 is under the control of ATRA inducible promoters.” This means they have RAREs?  Should be more specific.

Response: We specified: The difference in the A regions contributes to different transcriptional regulation in a cell promoter-specific and ligand-independent manner [9]. Expression of the isoforms RARα2, RARβ2, and RARɣ2 is induced by ATRA and is controlled by the promoters containing RAREs (retinoic acid response elements) sequences.

Line 66/67 The line “RARs dependent transcription activation is intrinsically linked to their proteasome-mediated degradation however, at the same time, ATRA upregulates the expression of RARa2, RARb2, and RARg2 isoforms.” Is very disjointed.  What does their  proteasome-mediated degradation have to do with ATRA regulation of receptors?  In what way is proteasome-mediated degradation important?  How does ATRA regulate the receptors?  Greatly lacking in explanation.  A few lines down please explain how this “maintains gene regulation and physiological effects of ATRA over an extended time”

Response: We changed the sentence: RARs dependent transcription activation is intrinsically linked to their proteasome-mediated degradation. At the same time, ATRA upregulates the expression of RARα2, RARβ2, and RARɣ2 isoforms, and induces proteasome-dependent degradation of retinoic acid receptor which is related to his ubiquitination as described for many other proteins. Therefore it might be possible that activation of transcription by RARs related to proteasome-mediated degradation and upregulation of RARs expression by ATRA was evolutionarily favored to sustain the expression of a given receptor and maintain gene regulation and physiological effects of ATRA over an extended time [10].

Line 82/83  “ATRA must be transported by the  cellular retinoic acid-binding protein (CRABP).”  Statement “must” Is too strong.  Mouse knockouts of CRABPI and II do not have strong effects on physiology.  Further discussion of CRABPs and which are expressed in skin is relevant and should be expanded.  However, a few sentences down this section jumps back to RAR function and function of helix 12 which has already been discussed.  This is quite disorganized.   An in depth discussion of RAR function is not needed and there are many good reviews on this already. 

Response: We rearranged the article and rewritten the paragraph to fix the issues. “Physiologically, to interact with its nuclear receptor, ATRA is transported into the nucleus by the CRABP that binds ATRA with high affinity. CRABP are divided into two subtypes CRABP I, and CRABP II that is more abundant in the skin [11]”

Line 94/95 “the broad range of ATRA effects is not induced solely by the RAR/RXR mechanism of action but also by the physiological effects of activation of any RXR heterodimerization partners [2].” Is incorrect.  Perhaps what is meant is that 9-cis RA activation of RXR may act on other partners of RXR?  This quite unclear as 9-cis RA is not mentioned.

Response: We have rewritten the paragraph: Therefore, the broad range of RA effects (through the 9-cis RA) is not induced solely by the RAR/RXR mechanism of action but also by the physiological effects of activation of any RXR heterodimerization partners [7]. ATRA and its isomers, 9-cis-RA and 13-cis-RA are the naturally occurring ligands for the RARs. They can bind three RARs with different affinities. The 9-cis-RA is a high-affinity ligand for RXR, and a selective agonist of RARα, RARβ, and RARɣ. The RARα is bound with the highest affinity by the 9-cis-RA followed by the 13-cis-RA and ATRA whereas for RARβ and RARɣ the order of affinity is reversed  [15,16]

Line 230/1 “significant interaction between ATRA and E3 Nrf2 (nuclear factor erythroid 2–related 230 factor 2).” What is meant by interaction?  They interact with each other?  Or does ATRA induce E3 Nrf2?  When Nrf2 is mentioned later is this the same as E3 Nrf2 or different?

Response: We changed the paragraph to: Cheng et al. (2019) [49] concluded that ATRA significant induces Nrf2 (nuclear factor erythroid 2–related factor 2). E3 ligase Hrd1 induces Nrf2 ubiquitylation and degradation in embryonic fibroblast cells in response to oxidative stress, while Nrf2 acts as an ATRA-activated photoprotective agent in the skin regeneration. In the mentioned research, ATRA reversed upregulation of E3 ligase Hrd1 expression and upregulation of Nrf2 expression in UV exposed cells both in vivo and in vitro [48].

Line 238/9 Can there be more explanation of the statement “RAR agonists stimulated type I procollagen protein expression and inhibited RAR-mediated MMP-3 and MMP-13 protein expression” It seems very odd that a RAR agonist would inhibit RAR-mediated protein expression.  Further, what actually is RAR-mediated protein expression?

Response: We corrected the sentence: . ATRA and RAR agonists stimulated type I procollagen protein expression and inhibited matrix metalloproteinases (MMPs) function. Inhibition occurred by downregulation of c-Jun protein and led to decreased collagen degradation in photoaged skin.

Line 250 The paragraph beginning “RAR-mediated MMP-3 and MMP-13 protein expression…” – is this describing that ATRA has a positive or negative action on skin.  What is the bottom line of this paragraph?

Response: The paragraph was rephased for clarification.

For figure 3 why have green arrow up and red down?  The direction of the arrow is enough.  Much more useful would be to e.g. use colour to describe whether this is a positive or negative effect on the skin in disease.  In addition, the actions of ATRA are complicated – does it always induce inflammation, as shown in the figure?  Or always decrease vascular remodelling?

Response: The figure have been updated. Actions described in the figure are induced by ATRA in majority of cases, however we do not exclude the possibility that ATRA may influence the skin in different way in specific cases.

Grammar needs significant improvement and I would suggest that this is given a thorough check for mistakes or lack of clarity.  To pick up some of these: Line 30/31       “Receptors” should not be plural

Response:  The manuscript was revised by the native English speaker.

Line 31-33       “Complex studies showed that temporary inactivation of those receptors in skin cells indicate greater applications of retinoic acid and its derivatives in a variety of serious dermatological disorders including some malignant conditions.” I don’t understand what is meant by this sentence.  What is meant by “greater applications”. Greater than what?  Also what is meant by complex studies?

Response:  The sentence was changed for clarification: The studies using complex genetic models with various combinations of RARs and RXRs  indicate, that retinoic acid and its derivatives have therapeutic potential for a variety of serious dermatological disorders including some malignant conditions.

Line 47            key developmentAL regulators

Line 53            mediator complexes should be mediator complex

Line 131          These strategies ARE BASED on modulation and increasing

Line 139          do not form characteristic patternS

Line 141          retinoiC acid

Response:  The issues were corrected.

Line 152/3       “(48 hours and 72 hours after exposition respectively)” what is meant by exposition?  Treatment with retinoic acid?

Response: Yes, the sentence was changed for clarification: It was shown that retinoic acid downregulates the expression of RARɣ and upregulates the expression of RARβ (48 hours and 72 hours after ATRA exposure, respectively) while RARα is not regulated by ATRA in vitro [25]

Line 157/8       “used in the treatment of psoriasis which allows for a better understanding of ATRAs role in the skin.” A better understanding than what method?

Response: The sentence was changed: Retinoids have been widely used in the treatment of psoriasis. This allows for a deeper understanding of the role of RAs in skin physiology and pathology

Line 168          “the onset demand for vitamin A is increased” what is meant by onset?

Response:  It was a editorial mistake. I was supposed to be: disease onset the demand. The sentence was changed for clarity. Wang et al. (2020) [54] have shown in psoriasis mouse models that during the disease the demand for vitamin A is increased and that the transformation from retinol to RA is also accelerated.

Line 183/4       “The fact that RA does not regulate RARα in normal skin but 183 does in psoriasis further indicates that RA has a “normalizing” effect on the gene expression.” Indicates is too strong and this is not sufficient to demonstrate this.  Perhaps suggests?

Response:  The sentence was changed: The fact that RA does not regulate RARα in normal skin but does in psoriasis further suggests that RA has a “normalizing” effect on the gene expression.

Line 281          “ATRA-containing medicines dedicated to topical administration” what is meant by dedicated?  Simply that it is used topically?

Response: The sentence was simplified: One of the ATRA-containing medicines for topical administration is a hydrophilic polymer PVA covalently linked by hydrolytically degradable linkage to ATRA (PATRA).

Line 307/8       “ATRA treatment resulted IN A decrease in the contents of wax esters, fatty acids, free fatty acids production. It indicated that ATRA APPLIED topically”

Response:  The sentence was corrected.

Line 313          “ Combinatory treatment with ATRA and other drugs brought out to be more effective than one drug application ordinary methods” I dont understand what is meant by ordinary methods.  A single drug?

Response:  Yes, the sentence was corrected: Combinatory treatment with ATRA and other drugs is more effective than one drug administration, yet despite the lesser toxicity of nano-ATRA, combinational therapy should be customized to a patient’s needs.

Line 330-33     The statement “The results did not suggest an increased risk of retinoid embryopathy. Moreover, topical retinoid therapy cannot be advised during pregnancy due to the possible negative effects on fetus development.” Is confusing as the first part contradicts the second – first saying studies show no teratogenicity and then immediately after saying it possibly is a teratogen.

Response: The paragraph was rewritten:
The results did not suggest an increased risk of retinoid embryopathy. However, based on the current state of knowledge topical retinoid therapy cannot be advised during pregnancy due to the possible negative effects on fetus development. Taken together, it shows that there are still a lot of gaps in the knowledge in understanding the role of retinoids in skin diseases. Future studies are necessary to complete this knowledge.

Reviewer 2 Report

Dear authors, after the reading of your work, I report some notes to improve manuscript:

  • “Here we provide a synopsis of the main advances in understanding the rol e of ATRA and its receptors in dermatology.” In abstract, focus of your article is more clear, but not reflect the title. I suggest changing title.
  • Manuscript’s title is “retinoic acid in dermatology”. From the introduction, you cite retinol and ATRA, but no retinoic acid. You must report in introduction a clear definition of retinol, retinoic acid and ATRA, explain pathways. Moreover, no reference are presented in introduction. Please, study and cite:
  • Khalil S, Bardawil T, Stephan C, Darwiche N, Abbas O, Kibbi AG, Nemer G, Kurban M. Retinoids: a journey from the molecular structures and mechanisms of action to clinical uses in dermatology and adverse effects. J Dermatolog Treat. 2017 Dec;28(8):684-696. doi: 10.1080/09546634.2017.1309349. Epub 2017 Apr 2. PMID: 28318351.
  • Niederreither, K., Dollé, P. Retinoic acid in development: towards an integrated view. Nat Rev Genet 9, 541–553 (2008). https://doi.org/10.1038/nrg2340
  • Ghyselinck NB, Duester G. Retinoic acid signaling pathways. Development. 2019 Jul 4;146(13):dev167502. doi: 10.1242/dev.167502. PMID: 31273085; PMCID: PMC6633611. To improve background.
  • “Signalling”, please
  • In vivo, in vitro, please.
  • “Retinoic acid in the skin”, maybe retinoic acid in dermatology disease? Then you talk about RAMBAs: no connection between title and arguments.
  • Paragraph 4, line 131-133, rephrase, please.
  • Manuscript’s title is “retinoic acid in Dermatology”. In all text, molecular aspect of ATRA have been reported, no reviews on single dermatologic disease, no focus on retinoic acid but on ATRA.
  • “Funding: National Science Center project OPUS17 number 2019/33/B/NZ5/02399 "Epigenetic effect on therapeutic activities of all-trans-retinoic acid in acute myeloid leukemia".” You should clarify the translational research and how from epigenetic effect of ATRA in AML you have been made a review of RA in Dermatology.
  • Paragraph 4, studies about ATRA-4-hydroxylase enzymes are in vitro, specify. Then you must improve background and cite other studies as:
  • Verfaille CJ, Borgers M, van Steensel MA. Retinoic acid metabolism blocking agents (RAMBAs): a new paradigm in the treatment of hyperkeratotic disorders. J Dtsch Dermatol Ges. 2008 May;6(5):355-64. English, German. doi: 10.1111/j.1610-0387.2007.06541.x. Epub 2007 Oct 16. Erratum in: J Dtsch Dermatol Ges. 2008 Jul;6(7):610. PMID: 17941881.
  • Purushottamachar P, Patel JB, Gediya LK, Clement OO, Njar VC. First chemical feature-based pharmacophore modeling of potent retinoidal retinoic acid metabolism blocking agents (RAMBAs): identification of novel RAMBA scaffolds. Eur J Med Chem. 2012 Jan;47(1):412-23. doi: 10.1016/j.ejmech.2011.11.010. Epub 2011 Nov 17. PMID: 22130607; PMCID: PMC3259215.
  • Paragraph 6. You were talking about a natural retinoid (RA by title), and now you talk about a synthetic retinoid. Then, “…by promoting the migration of keratinocytes in the epidermis and thus improves epidermal barrier function”. The study of Lee et al. is on Human Skin Equivalents, report, please.
  • Figure 2, non-genomic, please.
  • You had to re-organized the paper, and clarify if you are made an article on skin disease and the usage of retinoic acid, or others, it’s not well-defined.
  • “RARa”, RARα, please. Then, RARβ and γ.
  • Paragraph 7, line 141 “retinoic acid/vitamin A”. Please, study vitamin A pathways in Eukaryotes and human cell.
  • No research strategy is present. How did you chose articles or trials? Please, insert a section about material and method where elucidate you strategy and criteria.
  • In paragraph “Skin differentiation”, you talk about psoriasis. What is the rational?
  • “Many therapies have been proven to be effective in the treatment of skin conditions derived from the depletion of endogenous ATRA concentration.” It’s a leitmotif of all the manuscript. I remember you at first, in healthy human dosage of ATRA is not easy both for Analytic analyses both for hospital or University center. Despite you can measure ATRA serum level in AML patients, standard research is about retinol and BRP in serum, not ATRA.
  • To elucidate and improve molecular aspect in specific skin diseases refer to articles as:
  • Elena Doldo, Gaetana Costanza, Sara Agostinelli, Chiara Tarquini, Amedeo Ferlosio, Gaetano Arcuri, Daniela Passeri, Maria Giovanna Scioli, Augusto Orlandi, "Vitamin A, Cancer Treatment and Prevention: The New Role of Cellular Retinol Binding Proteins", BioMed Research International, vol. 2015, Article ID 624627, 14 pages, 2015. https://doi.org/10.1155/2015/624627
  • Passeri D, Doldo E, Tarquini C, Costanza G, Mazzaglia D, Agostinelli S, Campione E, Di Stefani A, Giunta A, Bianchi L, Orlandi A. Loss of CRABP-II Characterizes Human Skin Poorly Differentiated Squamous Cell Carcinomas and Favors DMBA/TPA-Induced Carcinogenesis. J Invest Dermatol. 2016 Jun;136(6):1255-1266. doi: 10.1016/j.jid.2016.01.039. Epub 2016 Mar 2. PMID: 26945879.
  • Campione E, Cosio T, Lanna C, Mazzilli S, Ventura A, Dika E, Gaziano R, Dattola A, Candi E, Bianchi L. Predictive role of vitamin A serum concentration in psoriatic patients treated with IL-17 inhibitors to prevent skin and systemic fungal infections. J Pharmacol Sci. 2020 Sep;144(1):52-56. doi: 10.1016/j.jphs.2020.06.003. Epub 2020 Jun 11. PMID: 32565006.
  • Sumita JM, Miot HA, Soares JLM, Raminelli ACP, Pereira SM, Ogawa MM, Picosse FR, Guadanhim LRS, Enokihara MMSS, Leonardi GR, Bagatin E. Tretinoin (0.05% cream vs. 5% peel) for photoaging and field cancerization of the forearms: randomized, evaluator-blinded, clinical trial. J Eur Acad Dermatol Venereol. 2018 Oct;32(10):1819-1826. doi: 10.1111/jdv.15020. Epub 2018 May 27. PMID: 29704456.
  • Campione E, Cosio T, Lanna C, Mazzilli S, Dika E, Bianchi L. Clinical efficacy and reflectance confocal microscopy monitoring in moderate-severe skin aging treated with a polyvinyl gel containing retinoic and glycolic acid: An assessor-blinded 1-month study proof-of-concept trial. J Cosmet Dermatol. 2020 May 1. doi: 10.1111/jocd.13463. Epub ahead of print. PMID: 32356917.
  • di Masi A, Leboffe L, De Marinis E, Pagano F, Cicconi L, Rochette-Egly C, Lo-Coco F, Ascenzi P, Nervi C. Retinoic acid receptors: from molecular mechanisms to cancer therapy. Mol Aspects Med. 2015 Feb;41:1-115. doi: 10.1016/j.mam.2014.12.003. Epub 2014 Dec 25. PMID: 25543955.

But explain further rational for research.

You have to re-organized your manuscript with material and methods( research strategy with inclusion and exclusion criteria) and focus on molecular pathways of retinoic acid in dermatology disease ( the journal is Cells, and scope “Cells covers every topic related to cell biology and physiology, molecular biology, and biophysics. Thus, our major focus is on experimental cytology rather than on clinical and epidemiological studies.”).

Author Response

Response to the Reviews

We want to thank the reviewers for all their valuable comments that allowed us to improve our manuscript. All changes of earlier text are marked in red and are included the current manuscript text according to all reviewer suggestions.  Below, we include also all reviewers’ comments with our responses.

First round  

Review 2:

“Here we provide a synopsis of the main advances in understanding the role of ATRA and its receptors in dermatology.” In abstract, focus of your article is more clear, but not reflect the title. I suggest changing title. Manuscript’s title is “retinoic acid in dermatology”. From the introduction, you cite retinol and ATRA, but no retinoic acid. You must report in introduction a clear definition of retinol, retinoic acid and ATRA, explain pathways.

Response:  We changed the title and included all necessary information in introduction.

Moreover, no reference are presented in introduction. Please, study and cite:

Khalil S, Bardawil T, Stephan C, Darwiche N, Abbas O, Kibbi AG, Nemer G, Kurban M. Retinoids: a journey from the molecular structures and mechanisms of action to clinical uses in dermatology and adverse effects. J Dermatolog Treat. 2017 Dec;28(8):684-696. doi: 10.1080/09546634.2017.1309349. Epub 2017 Apr 2. PMID: 28318351.

Niederreither, K., Dollé, P. Retinoic acid in development: towards an integrated view. Nat Rev Genet 9, 541–553 (2008). https://doi.org/10.1038/nrg2340

Ghyselinck NB, Duester G. Retinoic acid signaling pathways. Development. 2019 Jul 4;146(13):dev167502. doi: 10.1242/dev.167502. PMID: 31273085; PMCID: PMC6633611.

To improve background.

Response:  We improved the introduction and cited appropriate articles.

 “Signalling”, please, In vivo, in vitro, please.

Response:  We changed that.

“Retinoic acid in the skin”, maybe retinoic acid in dermatology disease? Then you talk about RAMBAs: no connection between title and arguments.

Response:  We have rearranged the article.

Paragraph 4, line 131-133, rephrase, please.

Response:  The paragraph was rewritten.

Manuscript’s title is “retinoic acid in Dermatology”. In all text, molecular aspect of ATRA have been reported, no reviews on single dermatologic disease, no focus on retinoic acid but on ATRA.

Response:  We changed the title and rephrased some paragraphs.

 “Funding: National Science Center project OPUS17 number 2019/33/B/NZ5/02399 "Epigenetic effect on therapeutic activities of all-trans-retinoic acid in acute myeloid leukemia".” You should clarify the translational research and how from epigenetic effect of ATRA in AML you have been made a review of RA in Dermatology.

Response:  The grant is mentioned because some of the authors’ salaries are supported by the grant.

Paragraph 4, studies about ATRA-4-hydroxylase enzymes are in vitro, specify. Then you must improve background and cite other studies as:

Verfaille CJ, Borgers M, van Steensel MA. Retinoic acid metabolism blocking agents (RAMBAs): a new paradigm in the treatment of hyperkeratotic disorders. J Dtsch Dermatol Ges. 2008 May;6(5):355-64. English, German. doi: 10.1111/j.1610-0387.2007.06541.x. Epub 2007 Oct 16. Erratum in: J Dtsch Dermatol Ges. 2008 Jul;6(7):610. PMID: 17941881.

Purushottamachar P, Patel JB, Gediya LK, Clement OO, Njar VC. First chemical feature-based pharmacophore modeling of potent retinoidal retinoic acid metabolism blocking agents (RAMBAs): identification of novel RAMBA scaffolds. Eur J Med Chem. 2012 Jan;47(1):412-23. doi: 10.1016/j.ejmech.2011.11.010. Epub 2011 Nov 17. PMID: 22130607; PMCID: PMC3259215.

Response:  The studies have been studied and cited in the text.

Paragraph 6. You were talking about a natural retinoid (RA by title), and now you talk about a synthetic retinoid. Then, “…by promoting the migration of keratinocytes in the epidermis and thus improves epidermal barrier function”. The study of Lee et al. is on Human Skin Equivalents, report, please.

Response: Missing information was reported in the manuscript.

Figure 2, non-genomic, please.

You had to re-organized the paper, and clarify if you are made an article on skin disease and the usage of retinoic acid, or others, it’s not well-defined.

“RARa”, RARα, please. Then, RARβ and γ.

Response:  The issues were corrected throughout the text.

Paragraph 7, line 141 “retinoic acid/vitamin A”. Please, study vitamin A pathways in Eukaryotes and human cell.

Response:  The issue was corrected.

No research strategy is present. How did you chose articles or trials? Please, insert a section about material and method where elucidate you strategy and criteria.

Response:  This review article is not a PRISMA style systemic review and therefore we believe that the material and method paragraph is not needed.

In paragraph “Skin differentiation”, you talk about psoriasis. What is the rational?

Response:  The text in the manuscript was re-organized.

 “Many therapies have been proven to be effective in the treatment of skin conditions derived from the depletion of endogenous ATRA concentration.” It’s a leitmotif of all the manuscript. I remember you at first, in healthy human dosage of ATRA is not easy both for Analytic analyses both for hospital or University center. Despite you can measure ATRA serum level in AML patients, standard research is about retinol and BRP in serum, not ATRA.

Response:  We changed ATRA to retinoids.

To elucidate and improve molecular aspect in specific skin diseases refer to articles as:

Elena Doldo, Gaetana Costanza, Sara Agostinelli, Chiara Tarquini, Amedeo Ferlosio, Gaetano Arcuri, Daniela Passeri, Maria Giovanna Scioli, Augusto Orlandi, "Vitamin A, Cancer Treatment and Prevention: The New Role of Cellular Retinol Binding Proteins", BioMed Research International, vol. 2015, Article ID 624627, 14 pages, 2015. https://doi.org/10.1155/2015/624627

Passeri D, Doldo E, Tarquini C, Costanza G, Mazzaglia D, Agostinelli S, Campione E, Di Stefani A, Giunta A, Bianchi L, Orlandi A. Loss of CRABP-II Characterizes Human Skin Poorly Differentiated Squamous Cell Carcinomas and Favors DMBA/TPA-Induced Carcinogenesis. J Invest Dermatol. 2016 Jun;136(6):1255-1266. doi: 10.1016/j.jid.2016.01.039. Epub 2016 Mar 2. PMID: 26945879.

Campione E, Cosio T, Lanna C, Mazzilli S, Ventura A, Dika E, Gaziano R, Dattola A, Candi E, Bianchi L. Predictive role of vitamin A serum concentration in psoriatic patients treated with IL-17 inhibitors to prevent skin and systemic fungal infections. J Pharmacol Sci. 2020 Sep;144(1):52-56. doi: 10.1016/j.jphs.2020.06.003. Epub 2020 Jun 11. PMID: 32565006.

Sumita JM, Miot HA, Soares JLM, Raminelli ACP, Pereira SM, Ogawa MM, Picosse FR, Guadanhim LRS, Enokihara MMSS, Leonardi GR, Bagatin E. Tretinoin (0.05% cream vs. 5% peel) for photoaging and field cancerization of the forearms: randomized, evaluator-blinded, clinical trial. J Eur Acad Dermatol Venereol. 2018 Oct;32(10):1819-1826. doi: 10.1111/jdv.15020. Epub 2018 May 27. PMID: 29704456.

Campione E, Cosio T, Lanna C, Mazzilli S, Dika E, Bianchi L. Clinical efficacy and reflectance confocal microscopy monitoring in moderate-severe skin aging treated with a polyvinyl gel containing retinoic and glycolic acid: An assessor-blinded 1-month study proof-of-concept trial. J Cosmet Dermatol. 2020 May 1. doi: 10.1111/jocd.13463. Epub ahead of print. PMID: 32356917.

di Masi A, Leboffe L, De Marinis E, Pagano F, Cicconi L, Rochette-Egly C, Lo-Coco F, Ascenzi P, Nervi C. Retinoic acid receptors: from molecular mechanisms to cancer therapy. Mol Aspects Med. 2015 Feb;41:1-115. doi: 10.1016/j.mam.2014.12.003. Epub 2014 Dec 25. PMID: 25543955.

But explain further rational for research.

Response:  We studied the articles and cited them throughout the manuscript.

You have to re-organized your manuscript with material and methods( research strategy with inclusion and exclusion criteria) and focus on molecular pathways of retinoic acid in dermatology disease ( the journal is Cells, and scope “Cells covers every topic related to cell biology and physiology, molecular biology, and biophysics. Thus, our major focus is on experimental cytology rather than on clinical and epidemiological studies.”).

Response:  We have reorganized the manuscript and introduced changes where necessary. As stated before this review article is not a PRISMA style systemic review and therefore we believe that the material and method paragraph is not needed.

Reviewer 3 Report

The paper is a review about the state of art of retinoids in the skin. Is a fine review mainly on biological action on the skin. It is interesting more for biologists or basic scientists than dermatologists.

Author Response

Response to the Reviews

We want to thank the reviewers for all their valuable comments that allowed us to improve our manuscript. All changes of earlier text are marked in red and are included the current manuscript text according to all reviewer suggestions.  Below, we include also all reviewers’ comments with our responses.

First round  

Review 3:

The paper is a review about the state of art of retinoids in the skin. Is a fine review mainly on biological action on the skin. It is interesting more for biologists or basic scientists than dermatologists.

Response: Our assumption was to provide just such presentation of retinoids in the paper.